# CONFIGURATION-TO-PERFORMANCE SCALING LAW WITH NEURAL ANSATZ

## ABSTRACT

Researchers build scaling laws to forecast the training performance of expensive large-scale pre-training runs. To improve predictability across a broader set of hyperparameters and enable simpler tuning at scale, we propose learning a *Configuration-to-Performance Scaling Law* (CPL): a mapping from the *full training configuration* to training performance. Because no simple functional form can express this mapping, we parameterize it with a large language model (LLM), and fit it with diverse open-source pretraining logs across multiple sources, yielding a *Neural* Configuration-to-Performance Scaling Law (NCPL). NCPL accurately predicts how training configurations influence the final pretraining loss, achieving 20-40% lower prediction error than the configuration-agnostic Chinchilla law and generalizing to runs using up to $10\times$ more compute than any run in the training set. It further supports joint tuning of multiple hyperparameters with performance comparable to hyperparameter scaling law baselines. Finally, NCPL naturally and effectively extends to richer prediction targets such as loss-curve prediction.

## 1 INTRODUCTION

As pretraining large language models is extremely costly (Liu et al., 2024a; Kimi Team et al., 2025a), it is critical to have predictability of performance before executing the training run with the biggest models. People use training runs of smaller models to build a scaling law that maps the number of model parameters $N$ and the amount of data $D$ to the predicted pretraining loss (Kaplan et al., 2020; Hoffmann et al., 2022). This paper proposes to build a more comprehensive scaling law that maps the **full** training configuration C to performance metrics P, such as the final loss, which we call the *Configuration-to-Performance Scaling Law* (CPL). Whereas fitting classical scaling laws requires careful hyperparameter tuning for each data point (Hoffmann et al., 2022; Porian et al., 2024), CPL can leverage all available training logs, including those are suboptimally tuned.

A priori, building a CPL seems to be an overly ambitious goal. The relationship between the configuration and performance is difficult, if not impossible, to have a pre-specified functional form like power law. Thus, we propose to use a *neural ansatz*: a neural network that maps C to P , with parameters learned from data collected across many existing experiments. It may appear that we won't have sufficient data from expensive training runs to build such a CPL. Fortunately, open-source pretraining studies, such as Marin (Marin, 2025), Step Law (Li et al., 2025d) and OLMo (OLMo Team et al., 2024), have recently released much more diverse public pretraining data.

Encouragingly, we find that training CPL is already feasible with current open pretraining logs and foundation models. We finetune Qwen3-1.7B (Yang et al., 2025) to predict performance metrics, including the final pretraining loss and the full loss curve, on over 3,000 pretraining logs from open-source projects. NCPL generalizes to OOD runs that use up to $10\times$ more compute than any run in the training set (section 2). It improves over classical methods along multiple axes:

1. NCPL learns how configurations affect the final loss, achieving higher accuracy than the Chinchilla scaling law, which only takes $N$ and $D$ as inputs (Figure 1, Left).
2. NCPL supports joint tuning of multiple hyperparameters. When restricted to learning rate and batch size, NCPL matches the predictive performance of StepLaw (Li et al., 2025d) (Figure 1, Middle).
3. NCPL can also be extended to predict the full loss curve, not just the final loss. (Figure 1, Right).
4. NCPL qualitatively learns nuanced interactions between hyperparameters.

We defer the discussion of related work to section B.

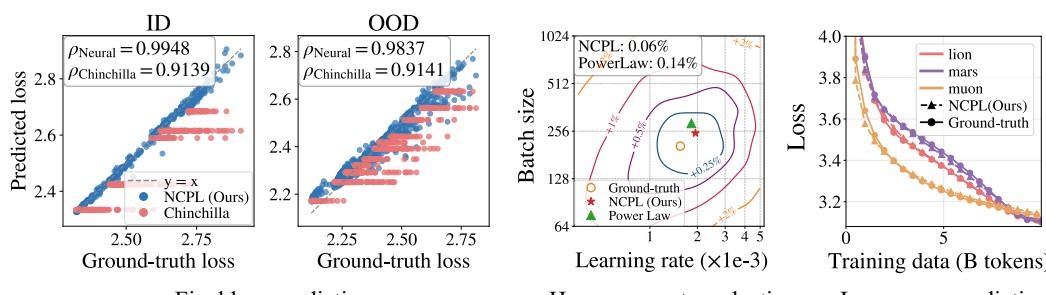

Final loss prediction       Hyperparameter selection       Loss curve prediction

Figure 1: **An Overview of NCPL's Performance Across Tasks. Left: NCPL predicts final loss more accurately than the Chinchilla Law.** Predicted vs. ground-truth loss on validation sets from the Steplaw dataset. The Chinchilla law yields configuration-agnostic prediction whereas NCPL takes in the full configuration as inputs and therefore achieves better prediction. **Middle: NCPL enables hyperparameter tuning.** Optimal learning rate and batch size prediction in an OOD setup (Steplaw dataset; $N = 536$M, $D = 28.4$B). Configuration-dependent predictions enable joint tuning over these two hyperparameters, achieving comparable performance to hand-designed functional form (Li et al., 2025d). **Right: NCPL can predict the entire loss curve beyond a single loss value.** Predicted vs. ground-truth pretraining loss curves under different optimizers on the validation sets (Marin dataset; $N = 520$M, $D = 10$B). NCPL predicts optimizer-specific curve shapes accurately.

## 2 METHODOLOGY

Large language model training outcomes are influenced by a wide range of factors, including model size $N$, data size $D$, model architecture, data recipe, optimizers, training hyperparameters, etc. We collectively refer to these factors as the training configuration C. We study how to learn a predictive model $f_\theta$ that maps configurations C to performance metrics P , such as the final pretraining loss (*Configuration–to-Performance Scaling Law (CPL)*). Pre-specifying a closed-form functional relationship for the entire high-dimensional and heterogeneous configuration space is extremely challenging, if not impossible, due to complex and nonlinear interactions among hyperparameters. Thus we parameterize the CPL using a generic neural network (specifically, a language model) and train it on a large collection of open-source pretraining runs (Hall et al., 2025; Li et al., 2025d). This yields what we refer to as the *Neural Configuration–Performance Scaling Law (NCPL)*.

**Input features and output metrics.** We use the training configuration of each pretraining run as the input features, with a full list provided in table 1. An example input instance is provided in fig. 5. For the main part of the paper, NCPL's target output is the final pretraining loss of each run. However, NCPL naturally extends to other objectives, and we present results in modeling the entire training loss curve in the last paragraph of section 3.1.

**Architecture.** We implement the regressor $f_\theta$ by finetuing a language model (Qwen3-1.7B(Yang et al., 2025)). Given the serialized input sequence $x$, the model embeds textual and numerical fields differently (fig. 5). Textual fields (including field names and categorical values such as optimizer type and learning-rate schedule) use the backbone language model's standard tokenizer and token embeddings. Numerical values (e.g., $N$, $D$, learning rate, and weight decay) are mapped to the model embedding space via a two-layer MLP. We obtain the scalar prediction by applying a linear layer to the last-layer hidden state at the last input position.

**Training.** Let $\mathcal{D}_{\text{train}} = \{(\text{C}^{(i)}, \text{P}^{(i)})\}_{i=1}^n$ denote a collection of pretraining runs, where $\text{C}^{(i)}$ is the training configuration of run $i$, and $\text{P}^{(i)} \in \mathbb{R}$ is the observed final pretraining loss (or an intermediate pretraining loss for the loss-curve prediction task, see section D for details). Rather than predicting the observed loss $\text{P}^{(i)}$ directly, we train the model to predict the residual relative to a Chinchilla-law baseline (Hoffmann et al., 2022). This design biases learning toward configuration-specific effects beyond the coarse dependence on model size and data size, and empirically improves extrapolation across scales (see ablations in section E.2). Concretely, we first fit a Chinchilla-law baseline $\hat{\ell}_{\text{chinchilla}}(N, D)$ on $\mathcal{D}_{\text{train}}$, which depends only on the number of model parameters $N$ and the number of training tokens $D$. For each run $i$, we define the residual regression target as

$$y^{(i)} = \text{P}^{(i)} - \hat{\ell}_{\text{chinchilla}}(N^{(i)}, D^{(i)}). \tag{1}$$

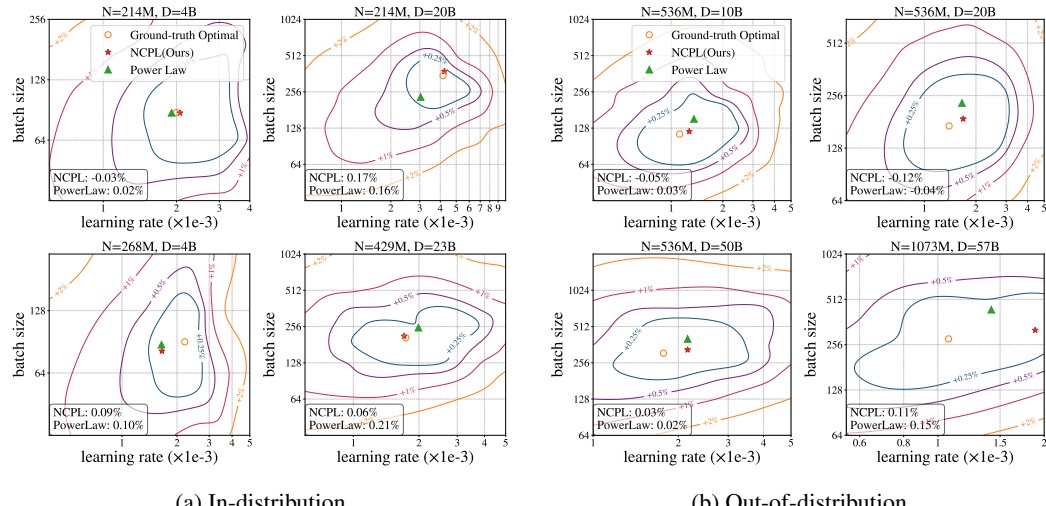

(a) In-distribution          (b) Out-of-distribution

Figure 2: **Hyperparameter selection with NCPL.** Predicted optimal learning rates and batch sizes from NCPL and the power-law fitting baseline (eq. (3)) on held-out ID and OOD $(N, D)$ pairs. The contour lines show the true loss landscape, with labels indicating relative losses to the minimum. The bottom legend reports the relative losses of the hyperparameters selected by NCPL and by the power law baseline. NCPL's prediction aligns with the true optima, and achieves comparable losses to the power-law baseline.

NCPL is trained by minimizing the mean squared error: $\mathcal{L}(\theta) = \sum_{i=1}^{n} \left( f_\theta(\mathrm{C}^{(i)}) - y^{(i)} \right)^2 / n$. At inference time, we recover the loss prediction by adding the baseline back.

To stabilize training, we adopt a two-stage fine-tuning scheme similar to the LP-FT method (Kumar et al., 2022): in Stage 1 updates only the two-layer MLP encoder for numerical fields and the linear prediction head, and Stage 2 fine-tunes all model parameters.

**Evaluation.** We report the performance of our NCPL and baselines on in-distribution (ID) and out-of-distribution (OOD) validation sets, where the OOD split consists of runs whose model sizes $N$ exceed those observed during NCPL training. The pretraining runs we collected roughly follow the Chinchilla scaling law, and the data-to-model ratio $N/D$ lies in the range $[1, 8]$. As a consequence, we are testing the extrapolation of NCPL to both larger model size and data size at the same time.

## 3 EXPERIMENTS

**Dataset.** We train and evaluate NCPL on pretraining logs from two open-source pretraining projects, from which we construct the *Marin Dataset* and the *Steplaw Dataset* of 2549 and 2581 training logs respectively. We designate all runs with model sizes larger than 430M parameters as an out-of-distribution (OOD) validation set. The remaining runs are split into training and in-distribution (ID) validation sets with an 8:2 ratio. Details are provided in section D.2.

**Model and fine-tuning details.** We use Qwen3-1.7B as the base model (Yang et al., 2025) and adopt the 2-stage LP-FT training described in Section 2. We find that training with multiple passes of data improves generalization, and therefore train for 20 epochs in Stage 1 and 1000 epochs in Stage 2 on the training set. More details are provided in section D.3.

### 3.1 MAIN RESULTS

**Configuration-dependent final loss prediction.** We first examine NCPL's ability to predict the final pretraining loss based on the full training configuration. In fig. 1 Left and fig. 6, each point shows the predicted versus ground-truth final pretraining loss for a single training run of a particular configuration. NCPL takes the full training configuration of each run as the input, whereas the Chinchilla Law baseline yields a single configuration-agnostic prediction for each model-data size pair. Consequently, NCPL aligns much more closely with the ground truth, achieving lower prediction errors and higher Spearman correlations on both ID and OOD validation sets, as summarized in table 2. Fig 7 and fig. 10 further visualizes fine-grained predictions over learning rate and batch size sweeps for held-out model-data size pairs from the StepLaw dataset. NCPL closely tracks

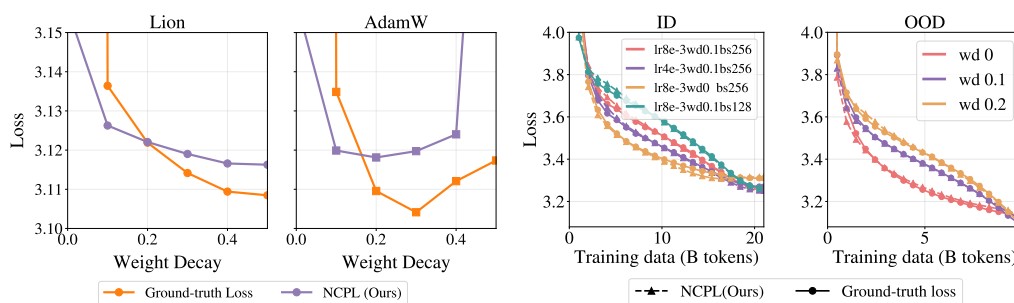

Figure 3: **NCPL learns interactions between weight decay and optimizer choice.** It's known that Lion requires substantially larger weight decay than AdamW (Chen et al., 2023; Loshchilov & Hutter, 2019; Wen et al., 2025). NCPL predicts this phenomenon on OOD validation sets (Marin, $N$=520M, $D$=10B).

Figure 4: **Loss curve prediction.** Ground-truth and predicted pretraining loss curves under different hyperparameter settings on ID and OOD validation sets. NCPL closely tracks the trajectories and learns hyperparameter-specific curve shapes. Settings provided in section D.5 Results under different optimizers are shown in figs. 1 and 13.

the ground-truth U-shaped loss curves across hyperparameters, illustrating its ability to learn how training hyperparameters affect the final loss.

**Hyperparameter selection.** With learned NCPL, one can sweep over training configurations and use NCPL's estimated loss as a proxy for the true pretraining loss, thereby identifying optimal hyperparameters without performing training sweeps (details provided in section D.4). In this section, we evaluate this capability on held-out model-data size pairs from the Steplaw dataset. We sweep over NCPL's predictions to select the optimal learning rate and batch size, and compare its predictions with the power-law scaling rule proposed in Li et al. (2025d) (eq. (3)). As shown in fig. 2, hyperparameters predicted by NCPL closely match the true optima obtained from exhaustive sweeps in both ID and OOD settings. The prediction error is comparable to that of fitted power-law baseline.

**Characterizing interactions between training configurations.** Different hyperparameters in the training configuration do not affect final pretraining loss independently; instead, their effects can interact in complicated ways. For example, different optimizers often prefer different hyperparameter choices (e.g., Liu et al. (2025b); Wen et al. (2025); Marek et al. (2025)). By learning a mapping from full training configurations to pretraining performance, we expect NCPL to learn such interactions from large-scale pretraining logs. Here we illustrate this potential with a concrete example. The Lion optimizer (Chen et al., 2023) is known to require substantially larger weight decay than AdamW (Loshchilov & Hutter, 2019; Wen et al., 2025). As shown in fig. 3, when using larger weight decay (e.g., $\geq 0.4$), AdamW's performance deteriorates markedly, whereas Lion improves. NCPL learns this interaction from the training set and generalizes it to the OOD setting.

**NCPL for loss curve prediction.** Beyond accepting the full training as inputs, the generality of NCPL also allows us to target other metrics beyond final pretraining loss. We train a variant of NCPL to perform loss curve prediction, and fig. 4 demonstrates the true loss curves and NCPL's predictions from the Marin Dataset. NCPL accurately predicts the loss curve under different hyperparameter choices on both ID and OOD generalization settings. This task previously required hand-designing complex multi-component power laws (Luo et al., 2025; Tissue et al., 2024), and we show that the shapes of loss curves can be learned directly from data. This also resonates with the recent finding on scaling collapse (Qiu et al., 2025), which suggests that the shapes of loss curves are the same across scales up to an affine transformation. Moreover, NCPL can also predict loss curves for different optimizers; see fig. 1 and fig. 13. Additional results and ablation study are provided in section E

## 4 CONCLUSION

In this work, we propose to learn the mapping from full training configurations to training outcomes using generic neural networks, by training on large-scale and diverse open-source pretraining logs. We instantiate this idea by fine-tuning a pretrained language model (Qwen3-1.7B (Yang et al., 2025)), resulting in the *Neural Configuration–Performance Scaling Law* (NCPL). Empirically, NCPL accurately predicts how configuration choices affect final pretraining performance, supports joint hyperparameter tuning and loss-curve prediction, and qualitatively reveals interaction effects among hyperparameters. We provide further discussions and outlook in section C.

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

## A PRELIMINARIES

**Classical scaling laws.** Kaplan et al. (2020) observed that the pretraining loss of large language models decreases monotonically and in a predictable manner as the number of model parameters $N$ and the number of training tokens $D$ increase. More specifically, they proposed that the pretraining loss approximately follows a power-law scaling with respect to $N$ and $D$. Subsequently, Hoffmann et al. (2022) introduced a revised formulation, known as the *Chinchilla law*:

$$\ell_{\text{chinchilla}}(N, D) = E + \frac{A}{N^{\alpha}} + \frac{B}{D^{\beta}}. \tag{2}$$

A key ingredient for accurate scaling-law fitting is proper hyperparameter tuning for each pair $(N, D)$ used to fit the law.

However, the Chinchilla law itself does not provide guidance on how to tune hyperparameters during pretraining, nor does it predict the pretraining loss for suboptimal training configurations. In this work, we address this limitation by modeling the pretraining loss as a function of the full training configuration using a neural network.

**Hyperparameter scaling law.** Hyperparameter selection plays a crucial role in LLM pretraining. Recent work approaches this problem by fitting parametric functions that map training scale (model and data size) to the optimal choice of hyperparameters (Kaplan et al., 2020; DeepSeek-AI et al., 2024; Bjorck et al., 2025; Porian et al., 2024; Hu et al., 2024; Wang et al., 2024; Li et al., 2025d; Zhou et al., 2026). For instance, Li et al. (2025d) models the optimal learning rate and batch size as power-law functions of model size and data size (with batch size modeled as a function of data size):

$$\eta(N, D) = c\, N^{\alpha} D^{\beta}, \quad B(D) = d\, D^{\gamma}. \tag{3}$$

Such approaches rely on strong inductive assumptions about the functional form of the parametric scaling laws. In contrast, NCPL enables hyperparameter selection without specifying an explicit functional form a priori by modeling the pretraining loss from heterogeneous training logs in a data-driven manner (section D.4).

# B RELATED WORK

**Scaling law.** Kaplan et al. (2020); Hoffmann et al. (2022) showed the pretraining loss of the Transformer-based LLMs follow a power-law relation with the model size and data size. Subsequent work explores refined fitting protocols and functional forms (Porian et al., 2024; Li et al., 2025c; Caballero et al., 2023), different model architectures (Ludziejewski et al., 2024; Wang et al., 2024), incorporating other training hyperparameters such as learning rate (Tissue et al., 2024; Xie et al., 2024; Luo et al., 2025; Li et al., 2025a) and loss curve prediction (Luo et al., 2025; Tissue et al., 2024; Qiu et al., 2025). Scaling law is also applied to broader domains and settings, including multi-modal models (Henighan et al., 2020) , hyperparameter optimization (Kadra et al., 2023; Hashimoto, 2021), (Jain et al., 2024), adversarial attacks (Liu et al., 2024b), transfer learning(Hernandez et al., 2021), etc. Recent works extends scaling laws to model downstream tasks performance rather than the pretraining loss while some other works point out practical challenges (Gadre et al., 2025; Ruan et al., 2024; Bhagia et al., 2024; Khatri et al., 2025; Lourie et al., 2025) Scaling laws are also extended to predict the optimal hyperparameters with the scale of training resources (DeepSeek-AI et al., 2024; Li et al., 2025d; Bergsma et al., 2025; Bjorck et al., 2025; Marek et al., 2025; Li et al., 2025b; Zhou et al., 2026). However, these approaches fit only a small subset of hyperparameters, making it difficult to ensure that the remaining hyperparameters stay optimal during fitting or extrapolation; moreover, they do not model non-parametric configuration choices such as the optimizers.

Recently, Lin et al. (2025) proposes to use advanced LLMs in a scaffold to propose the functional form of scaling laws under different setups. However, identifying the correct functional form mapping from the *entire* configuration C to performance P can be extremely difficult due to the complicated interactions of different factors. We adopt a different methodology to directly use LLM as the regressor to predict the performance from training configurations.

**Theory-motivated studies of hyperparameter effects and transfer.** As complement to data-driven scaling-law fitting, an orthogonal approach studies the effect of hyperparameters and its transfer across training scales through theoretical lens. Using SDEs as a proxy for stochastic-gradient–based optimization, prior work has proposed scaling rules for how the learning rate should grow with batch size (Jastrzębski et al., 2017; Malladi et al., 2022). The critical batch size, beyond which large-batch training stops yielding proportional efficiency gains, can be characterized via the gradient-noise-scale proxy (McCandlish et al., 2018), and its scaling behavior has been further analyzed under simplified assumptions (Zhang et al., 2025). Wang & Aitchison (2025) interpret model weights as an exponential moving average of recent updates, and use this view to derive practical scaling rules for weight decay as model and dataset size grow. $\mu$P tackles how optimal learning rate can be transferred from smaller to larger models (Yang et al., 2021). Later works extended from the original width scaling setup to depth scaling and other architectural variants (Bordelon et al., 2024; Dey et al., 2025; Blake et al., 2025; Everett et al., 2024). However, $\mu P$ primarily focuses on model scaling and does not directly address learning-rate transfer under data scaling, and offers limited guidance for hyperparameters beyond the learning rate. Recent work argues weight decay plays an important role in stabilizing the update dynamics and thus facilitating learning rate transfer (Kosson et al., 2025), and propose joint transfer principle of learning rate and weight decay (Fan et al., 2025). While these works offer valuable insights into how hyperparameters influence pretraining performance and provide practical tuning and transfer guidelines, they typically address only a limited subset of hyperparameters and do not predict the training loss for a given configuration, as NCPL does.

**Foundation models as regressors.** Transformer-based foundation models have demonstrated strong capability in modeling complex input-output relationships beyond natural language processing. Recent work has shown that pretrained language models can be used as general-purpose regressors by direct prompting pretrained models (Vacareanu et al., 2024) or through supervised learning (Garg et al., 2022; Hollmann et al., 2022; Huang et al., 2020). Compared to classical regression approaches, foundation models offer several key advantages, including leveraging the semantic meaning of features to perform feature selection (Jeong et al., 2025), enabling large-scale pretraining on diverse data sources (Hollmann et al., 2025), and supporting online adaptation fine-tuning (Song et al., 2024). At the same time, recent studies have highlighted potential limitations, including sensitivity to data representation that are irrelevant to the underlying learning task (Liu et al., 2025a). Foundation models as regressors have been explored in a wide range of domains, for instance, time-

series forecasting (Gruver et al., 2023; Jin et al., 2024; Das et al., 2024; Goswami et al., 2024) and performance prediction of complex engineered systems (Akhauri et al., 2025). Our work focuses on predicting the pretraining outcomes of LLMs from full training configurations. To the best of our knowledge, this setting has not been systematically studied in prior work.

**Learning-curve extrapolation and hyperparameter optimization.** Beyond fitting parametric scaling laws, another line of work aims to *predict training trajectories* (learning curves) from partially-observed training trajectories, enabling early stopping and grey-box resource allocation in hyperparameter optimization. Early neural and probabilistic approaches include Bayesian neural networks with specialized learning-curve parameterizations (Klein et al., 2017) and probabilistic rollouts for learning-curve extrapolation across hyperparameter settings using models such as Bayesian RNNs (Gargiani et al., 2019). Ranking-based formulations further learn to rank partially observed curves to guide early termination (Wistuba & Pedapati, 2020). More recently, transformer-based Prior-Data Fitted Networks (PFNs) enable fast approximate Bayesian learning-curve extrapolation in a single forward pass (Adriaensen et al., 2023), and extend naturally to freeze-thaw Bayesian optimization via in-context surrogates (Rakotoarison et al., 2024). These PFN surrogates can also be specialized to optimizer hyperparameters, e.g., Adam tuning (Athanasiadis et al., 2025). In parallel, Deep Power Laws exploit power-law structure for grey-box HPO and budget allocation (Kadra et al., 2023). A very recent work Hu et al. (2026) tries to train a language model to predict downstream performance with the entire accuracy trajectories from a partial run, using token-level validation loss as input.

## C DISCUSSIONS

Despite these promising results, our current study should be viewed as a proof of concept due to the limited accessibility of open-source pretraining logs. In particular, our training dataset only includes models with up to 430M parameters and OOD validation set only includes models with at most 1.2B parameters. Moreover, the diversity of the configurations in the pretraining logs are limited. For example, in the open-source pretraining logs, for AdamW, hyperparameter $\beta_1$, $\beta_2$, and $\epsilon$ are rarely tuned, making it difficult to reliably learn their effects and interactions from the logs. Another example is that the pretraining logs do not have any MoE models or models with linear attention (Dai et al., 2024; Kimi Team et al., 2025b; Yang et al., 2024), and only contain two choices of pre-training datasets (Li et al., 2024)

Looking ahead, we expect NCPL-style predictors to improve with the community's collective efforts to open-source pretraining spanning diverse setups. As more public training runs are released, the community can build a shared NCPL from pooled data, and users can further fine-tune their own NCPL starting from this shared base. Looking ahead, an NCPL may ingest orders of magnitude more training runs than any individual human researcher, while possessing principled understanding comparable to that of a human researcher through the knowledge in the base model. While an NCPL likely cannot extrapolate to completely unknown scenarios such as runs testing a novel model architecture or a new dataset, one can continue training it with training logs data from the new scenarios, and expect some level of transfer based on the prior knowledge encoded in the network. This may be more data-efficient than the existing approach of building a brand-new scaling law with all new exper

Table 1: Training configuration fields used as inputs to NCPL. For numerical fields, we report the scaling factor applied before feeding them into the numerical encoder.

| Field group | Included fields | Field type | Scaling Factor |
|---|---|---|---|
| Source tag | Source identifier indicating which open-source training project the run comes from (e.g., Marin or StepLaw) | Categorical | – |
| Model architecture | Model size $N$ (number of non-embedding parameters, in millions) | Numerical | $0.01\times$ |
| | Number of layers | Numerical | $1\times$ |
| | Number of attention heads | Numerical | $1\times$ |
| | Hidden dimension | Numerical | $0.01\times$ |
| Training scale | Number of training tokens $D$ (in billions) | Numerical | $1\times$ |
| | Total number of training steps | Numerical | $0.001\times$ |
| | Current ratio of total training steps (for loss curve prediction) | Numerical | $1\times$ |
| Optimizer and hyperparameters | Optimizer | Categorical | – |
| | Peak learning rate | Numerical | $10^4\times$ |
| | Learning-rate schedule | Categorical | – |
| | Final learning rate after decay | Numerical | $10^4\times$ |
| | The ratio between final learning rate and peak learning rate | Numerical | $200\times$ |
| | Weight decay | Numerical | $10^2\times$ |
| | Batch size | Numerical | $10^{-1}\times$ |
| | Warmup ratio | Numerical | $10^{-2}\times$ |
| | Gradient clipping threshold | Numerical | $1\times$ |
| | Momentum coefficients $(\beta_1, \beta_2)$ for AdamW | Categorical | – |
| | Numerical-stability constant $\epsilon$ for AdamW | Categorical | – |
| | Adam learning rate used inside Muon optimizer | Numerical | $10^4\times$ |
| | Block size for the Soap optimizer | Numerical | $0.02\times$ |
| | Preconditioner learning rate for Kron optimizer | Categorical | – |

# D   EXPERIMENTAL DETAILS

## D.1   DATA PROCESSING

**Output Metrics**   Since the per-step pre-training loss exhibits high variance, we use less noisy metrics. For the Marin dataset, we use the language modeling loss on the English split of the C4 dataset (Raffel et al., 2020). For the StepLaw dataset, we use the exponentially smoothed training loss with a smoothing coefficient of 0.99.

**Filtering.**   The open-source runs we collect include diverged or failed runs, so we apply a filtering procedure with the following criteria: (i) unfinished runs; (ii) diverged runs, defined as those with final pretraining loss $> 4$, or with final loss more than 0.3 above the best loss achieved at the same data size and model size (a large gap in language-model pretraining); (iii) unstable runs, defined as those with an average loss slope larger than 0.001 over any 5% window of training steps.

**Training configuration used.**   In table 1, we list all training-configuration fields used as inputs to NCPL. Categorical fields are encoded using the backbone language model's standard tokenizer and token embeddings. Numerical values are mapped into the embedding space via a two-layer MLP. For $(\beta_1, \beta_2)$, $\epsilon$, and the preconditioner learning rate of the Kron optimizer, since these values lie in a small discrete range, we treat them as categorical and encode them with the standard tokenizer. For $\epsilon$, we additionally apply a negative $\log$ transform.

**Scaling.**   Numerical configuration values can vary by orders of magnitude, yet they are all processed by the same numerical encoder. We therefore apply a fixed scaling to each numerical field

when constructing inputs, so that different fields fall into a comparable range. The scaling factors are reported in table 1.

Example training samples for final-loss prediction and loss curve prediction are shown in fig. 5.

## D.2 DATASET

We train and evaluate NCPL on pretraining logs collected from two open-source pretraining projects, from which we construct the following two datasets:

1. **Marin Dataset.** Collected from the Marin Fantastic Optimizers Project (Hall et al., 2025; Wen et al., 2025), which systematically studies the performance and scalability of different optimizers for language model pretraining. The project conducts extensive hyperparameter sweeps across model sizes ranging from 130M to 1.2B parameters and data scales up to 193B tokens.[1]

2. **Steplaw Dataset.** Collected from the Steplaw Project (Li et al., 2025d). The project performs fine-grained sweeps over learning rate and batch size for models ranging from 215M to 1B parameters and data scales up to 100B tokens, while fixing all other hyperparameters and using the AdamW optimizer (Loshchilov & Hutter, 2019).[2]

To make the splits between training set and ID validation set more meaningful and challenging, we randomly split on the level of *(optimizer, model size, data size)* tuples. Concretely, we first group runs that share the same $(optimizer, N, D)$ tuple, and then randomly assign each group to either the training set or the ID validation set. This strategy makes sure that every run in the ID validation set (or the OOD validation set) does not have another run with the same optimizer, model size, data size in the training dataset. In total, the dataset comprises 3,225 training runs, 796 ID validation runs, and 1,109 OOD validation runs.[3]

## D.3 TRAINING SETUP AND HYPERPARAMETERS

As described in **??**, we adopt a two-stage training pipeline for stability. We use Qwen3-1.7B as the base model (Yang et al., 2025). In the first stage, we freeze the backbone and update only the two-layer MLP encoder for numerical fields together with the linear prediction head. We train for 20 epochs with learning rate $5 \times 10^{-5}$ and a warmup ratio of 0.1 of total steps. In the second stage, we fine-tune all model parameters for 1000 epochs using learning rate $1 \times 10^{-5}$ with a 1000-step warmup. We reset the optimizer state between the two stages. In both stages, we use AdamW with linear learning-rate decay, weight decay 0.01, and batch size 480.

**Training NCPL for loss curve prediction.** For each run we uniformly sample up to 40 intermediate checkpoints, append a scalar field indicating the fraction of total training steps completed, and train the model to predict the difference between the current loss and the Chinchilla baseline (fig. 5 Right). We use the same two-stage procedure and hyperparameters, but train for 10 epochs in the first stage and 400 epochs in the second stage.

**Chinchilla baseline fitting in residual prediction.** As described in section A, for each $(N, D)$, we select the run with the lowest final pretraining loss across different configurations, and fit the Chinchilla-law form (eq. (2)) to this collection of selected runs.[4] We emphasize that the baseline is fit using only the training set, so no information from validation runs is leaked.

## D.4 NCPL FOR HYPERPARAMETER SELECTION

NCPL predicts a loss value from the full training configuration. This enables selecting multiple hyperparameters jointly without running expensive training sweeps. Specifically, for a target model

---

[1]Pretraining logs are available at `https://wandb.ai/marin-community/optimizer-scaling`.

[2]Pretraining logs are available at `https://wandb.ai/billzid/predictable-scale`.

[3]The dataset and the training code are included in the Supplementary Material and will be open-sourced later.

[4]When training on multiple sources, we fit the baseline separately per source.

size $N$ and data size $D$, one can enumerate a discrete grid of candidate training configurations and select the configuration that minimizes NCPL's estimated final loss, which serves as a proxy for the true final pretraining loss. Experimental results are provided in section 3.1.

## D.5 EVALUATIONS

**Loss Curve Prediction** Setting of fig. 4: Marin Dataset. **Left:** ID validation, $N = 130M$, $D = 21B$, AdamW with varying learning rate, weight decay, and batch size. **Right:** OOD validation, $N = 520M$, $D = 10B$, Muon with learning rate $8 \times 10^{-3}$ and varying weight decay.

**Hyperparameter selection.** The contour plots are generated by interpolating the scattered ground-truth loss values in log-scaled learning-rate/batch-size space using a smooth RBF interpolant, followed by light Gaussian smoothing; we then draw iso-loss contour lines on the resulting surface. The "minimum" markers for both ground truth and NCPL are obtained by fitting a quadratic surface in log space using only near-optimal points whose loss is within $1\%$ of the minimum predicted loss, and taking the minimizer of the fitted quadratic.

# E ADDITIONAL RESULTS

## E.1 MORE RESULTS OF FINE-TUNED NCPL.

Final pretraining loss prediction results across learning rates and batch sizes on the Steplaw dataset are shown in fig. 7 (ID) and fig. 8 (OOD). Hyperparameter selection results for all held-out model–data size pairs are shown in fig. 9 and fig. 10. At the largest extrapolation scale ($N = 1073$M), NCPL's predicted pretraining loss tends to be higher than the ground-truth loss. This is partly because the Chinchilla baseline itself overpredicts at this scale; adding our predicted residual on top of an inflated baseline can further increase the final prediction. In addition, the model predicts a larger optimal learning rate for larger $N$ when the $D/N$ ratio is small. This trend may reflect limited data diversity in the training set, and could be alleviated by incorporating additional training logs. Loss-curve prediction results on the Marin dataset across different optimizers (ID and OOD) are presented in fig. 13. Fig. 11 and fig. 12 show randomly sampled loss curve prediction results on ID and OOD runs, without cherry-picking.

## E.2 ABLATION ON NCPL DESIGN CHOICES

We study two key design choices in NCPL: (i) predicting residuals relative to a Chinchilla-law baseline, and (ii) encoding scalar configuration values using numerical tokens via a two-layer MLP. We consider two ablated variants: replacing residual prediction with respect to the Chinchilla baseline with direct loss prediction, and replacing the two-layer MLP encoder with standard tokenization for numerical fields. Results shown in table 3 show consistent gains from both residual prediction and numerical tokens.

We study two key design choices in NCPL: (i) predicting residuals relative to a Chinchilla-law baseline, and (ii) encoding scalar configuration values using numerical tokens. Specifically, we consider two ablations: (a) predicting the final loss directly, rather than the residual w.r.t. the Chinchilla baseline; and (b) tokenizing numerical fields with a standard tokenizer, rather than encoding them with a two-layer MLP. As shown in table 3, both residual prediction and numerical-token encoding contribute substantially to performance.

## E.3 ADDITIONAL RESULTS OF NCPL WITH TRAINING FROM SCRATCH.

Fig. 14 and fig. 15 show the predicted vs. ground-truth final pretraining loss of NCPL trained from scratch, using 1.7B model and 135M model respectively.

## E.4 ABLATION ON THE BACKBONE ARCHITECTURE OF NCPL

In this section, we ablate the effect of the chosen backbone for NCPL, with the results shown in Table 2. We compare NCPL with XGBoost (Chen & Guestrin, 2016) and Chinchilla Law (Hoffmann et al., 2022) baselines on final-loss prediction and loss curve prediction tasks. For NCPL, we further ablate fine-tuning versus training from scratch with different model sizes, using the Qwen3-1.7B (Yang et al., 2025) and SmolLM2-135M (Allal et al., 2025) architectures.

Comparing with XGBoost and Chinchilla Law, NCPL achieves substantially lower prediction error and higher Spearman correlation than the configuration-agnostic Chinchilla-law baseline by leveraging the full training configuration, rather than only $(N, D)$. On the Steplaw dataset, where the configuration variation is restricted to a small set of continuous factors ($N$, $D$, learning rate, and batch size), training from scratch can outperform fine-tuning. In contrast, on the Marin dataset, where many heterogeneous fields vary jointly (e.g., optimizer choices and a wide range of hyperparameters), fine-tuning yields a clear advantage over both training from scratch and XGBoost. Therefore, we adopt the fine-tuned NCPL as our main method.

Table 2: Comparison of various versions of NCPL, XGBoost, and Chinchilla-law baselines for final-loss prediction and loss-curve prediction on both ID and OOD splits. For NCPL, we ablate fine-tuning (*ft*) versus training from scratch (*scratch*) with two backbone sizes (1.7B and 135M). We report mean absolute error (MAE), root mean squared error (RMSE), and Spearman correlation ($\rho$). Overall, NCPL achieves substantially lower error and higher rank correlation than the Chinchilla-law by leveraging full training configurations. On Steplaw dataset, where only a small set of hyper-parameters vary (learning rate and batch size), NCPL (training from scratch) are competitive with NCPL with fine-tuning; in contrast, on Marin dataset, which has more diverse configurations, NCPL with fine-tuning provides a clear advantage over other variants and baselines.

| Data | Method | Final-loss prediction | | | | | | Loss Curve Prediction | | | |
| | | In-distribution | | | Out-of-distribution | | | In-distribution | | Out-of-distribution | |
| | | MAE↓ | RMSE↓ | $\rho$↑ | MAE↓ | RMSE↓ | $\rho$↑ | MAE↓ | RMSE↓ | MAE↓ | RMSE↓ |
| | NCPL (ft, 1.7B) | **0.0109** | **0.0169** | **0.9813** | **0.0168** | **0.0239** | 0.9299 | **0.0252** | **0.0475** | **0.0363** | **0.0644** |
| | NCPL (scratch, 1.7B) | 0.0119 | 0.0171 | 0.9774 | 0.0207 | 0.0301 | 0.9324 | 0.0289 | 0.0554 | 0.0386 | 0.0707 |
| Marin | NCPL (scratch, 135M) | 0.0140 | 0.0211 | 0.9777 | 0.0266 | 0.0347 | **0.9421** | 0.0306 | 0.0527 | 0.0376 | 0.0713 |
| | XGBoost | 0.0188 | 0.0277 | 0.9754 | 0.0325 | 0.0375 | 0.9227 | 0.0305 | 0.0541 | 0.0385 | 0.0667 |
| | Chinchilla Law | 0.0566 | 0.0676 | 0.9004 | 0.0240 | 0.0326 | 0.7670 | – | – | – | – |
| | Per-optimizer Chinchilla | 0.0416 | 0.0575 | 0.8268 | 0.0211 | 0.0278 | 0.8062 | – | – | – | – |
| | NCPL (ft, 1.7B) | 0.0097 | 0.0158 | 0.9948 | 0.0223 | 0.0345 | 0.9837 | 0.0278 | 0.0981 | 0.0415 | 0.1696 |
| | NCPL (scratch, 1.7B) | 0.0090 | 0.0150 | **0.9956** | 0.0225 | 0.0313 | 0.9876 | 0.0258 | **0.0941** | 0.0384 | 0.1709 |
| Steplaw | NCPL (scratch, 135M) | **0.0082** | **0.0147** | 0.9953 | **0.0199** | **0.0284** | **0.9910** | **0.0265** | 0.0943 | 0.0376 | **0.1666** |
| | XGBoost | 0.0095 | 0.0162 | 0.9947 | 0.0246 | 0.0332 | 0.9863 | 0.0275 | 0.1068 | 0.0471 | 0.1834 |
| | Chinchilla Law | 0.0704 | 0.0949 | 0.9139 | 0.0440 | 0.0670 | 0.9141 | – | – | – | – |

Table 3: Ablations on final-loss prediction on both ID and OOD splits. We compare NCPL (Ours) with two ablations: removing residual prediction and removing numerical tokens. We report mean absolute error (MAE), root mean squared error (RMSE), and Spearman correlation ($\rho$).

| Data | Method | Final-loss prediction | | | | | |
| | | In-distribution | | | Out-of-distribution | | |
| | | MAE↓ | RMSE↓ | $\rho$↑ | MAE↓ | RMSE↓ | $\rho$↑ |
| | NCPL (Ours) | **0.0109** | **0.0169** | **0.9813** | **0.0168** | **0.0239** | **0.9299** |
| Marin | No Residual Prediction | 0.0133 | 0.0195 | 0.9745 | 0.1503 | 0.1576 | 0.9264 |
| | No Numerical Tokens | 0.0123 | 0.0243 | 0.9665 | 0.0217 | 0.0288 | 0.9267 |
| | NCPL (Ours) | **0.0097** | **0.0158** | 0.9948 | **0.0223** | **0.0345** | **0.9837** |
| Steplaw | No Residual Prediction | 0.0208 | 0.0301 | **0.9968** | 0.0988 | 0.1134 | 0.9607 |
| | No Numerical Tokens | 0.0123 | 0.0212 | 0.9906 | 0.0357 | 0.0428 | 0.9763 |

```
source:  steplaw
data size:  25.0
model size:  268.0
num layers:  8
num heads:  16
hidden dim:  9552
optimizer:  adamw
learning rate:  0.000977
lr schedule:  cosine
warmup:  2000
weight decay:  0.1
batch size:  960
beta1:  0.9
beta2:  0.95
epsilon:  8
max_grad_norm:  1.0
minlr ratio:  0.00102
minlr:  1e-05
max step:  127155
final loss:  0.0235
```

```
source:  marin
data size:  26.8
model size:  134.0
num layers:  32
num heads:  8
hidden dim:  512
optimizer:  soap
learning rate:  0.0016
lr schedule:  cosine
warmup:  1000
weight decay:  0.1
batch size:  1280
beta1:  0.95
beta2:  0.98
epsilon:  15
max_grad_norm:  1.0
minlr ratio:  0
minlr:  0
block_size:  256
max step:  127155
frac:  0.3907
final loss:  0.6609
```

Figure 5: Example training samples. Left: an input for final loss prediction. Right: an input for loss curve prediction. Numerical values are embedded with a two-layer MLP, while other text is embedded using standard token embeddings. The values 0.0235 and 0.6609 denote target labels and are not part of the input.

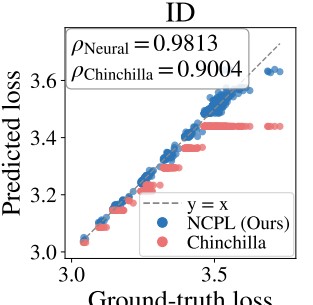
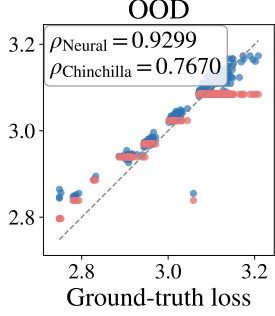

Figure 6: **Predicted loss vs. ground-truth loss.** Each point visualizes the predicted vs. ground-truth final pretraining loss of an individual run from the Marin dataset (for StepLaw, see fig. 1 left). NCPL uses the full training configuration as input, whereas the Chinchilla law only depends on $(N, D)$ and therefore gives the same prediction for all runs sharing the same $(N, D)$. As a result, NCPL achieves substantially higher Spearman correlation $\rho$.

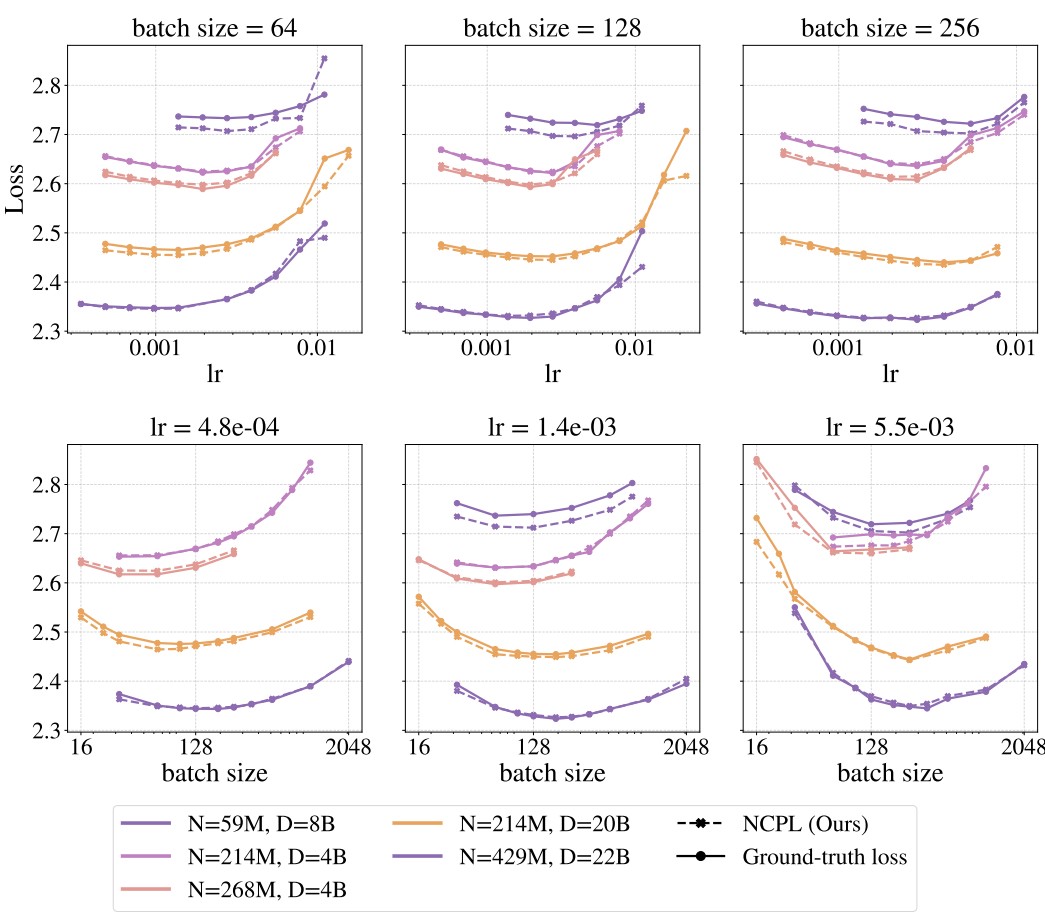

Figure 7: Final-loss prediction across learning rates and batch sizes for all 5 ID held-out $(N, D)$ pairs.

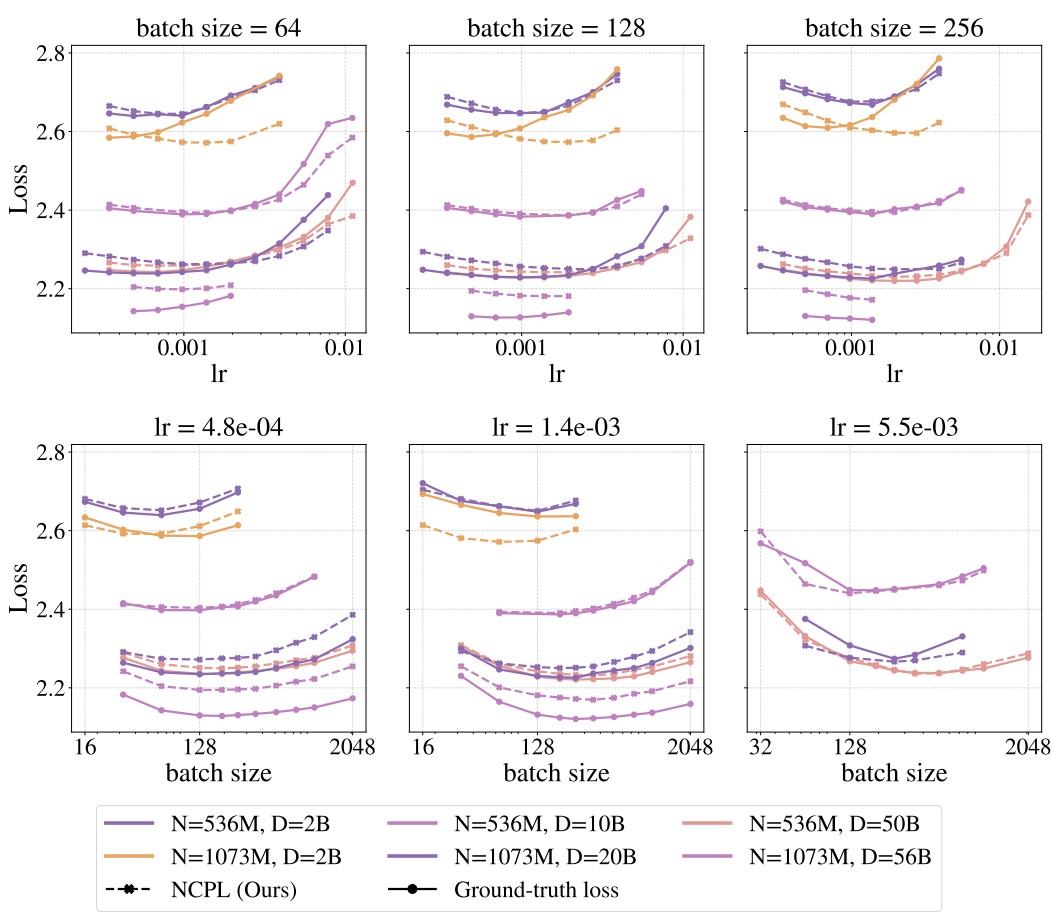

Figure 8: Final-loss prediction across learning rates and batch sizes for 6 OOD held-out $(N, D)$ pairs.

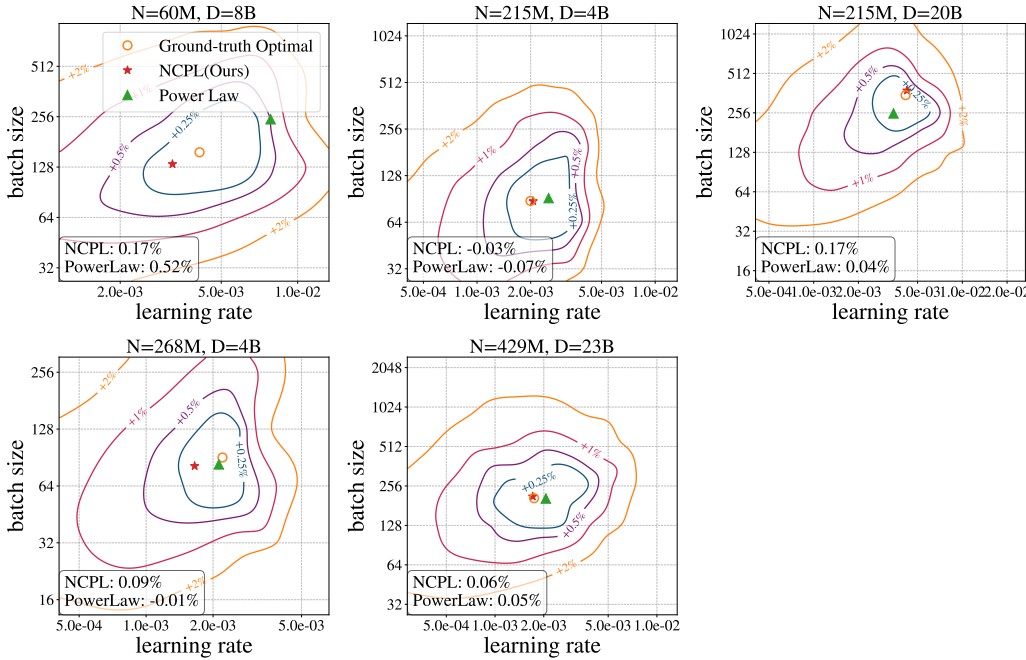

Figure 9: Predicted optimal learning rates and batch sizes for NCPL and the power law baseline on all 5 held-out ID $(N, D)$ pairs, together with their relative loss.

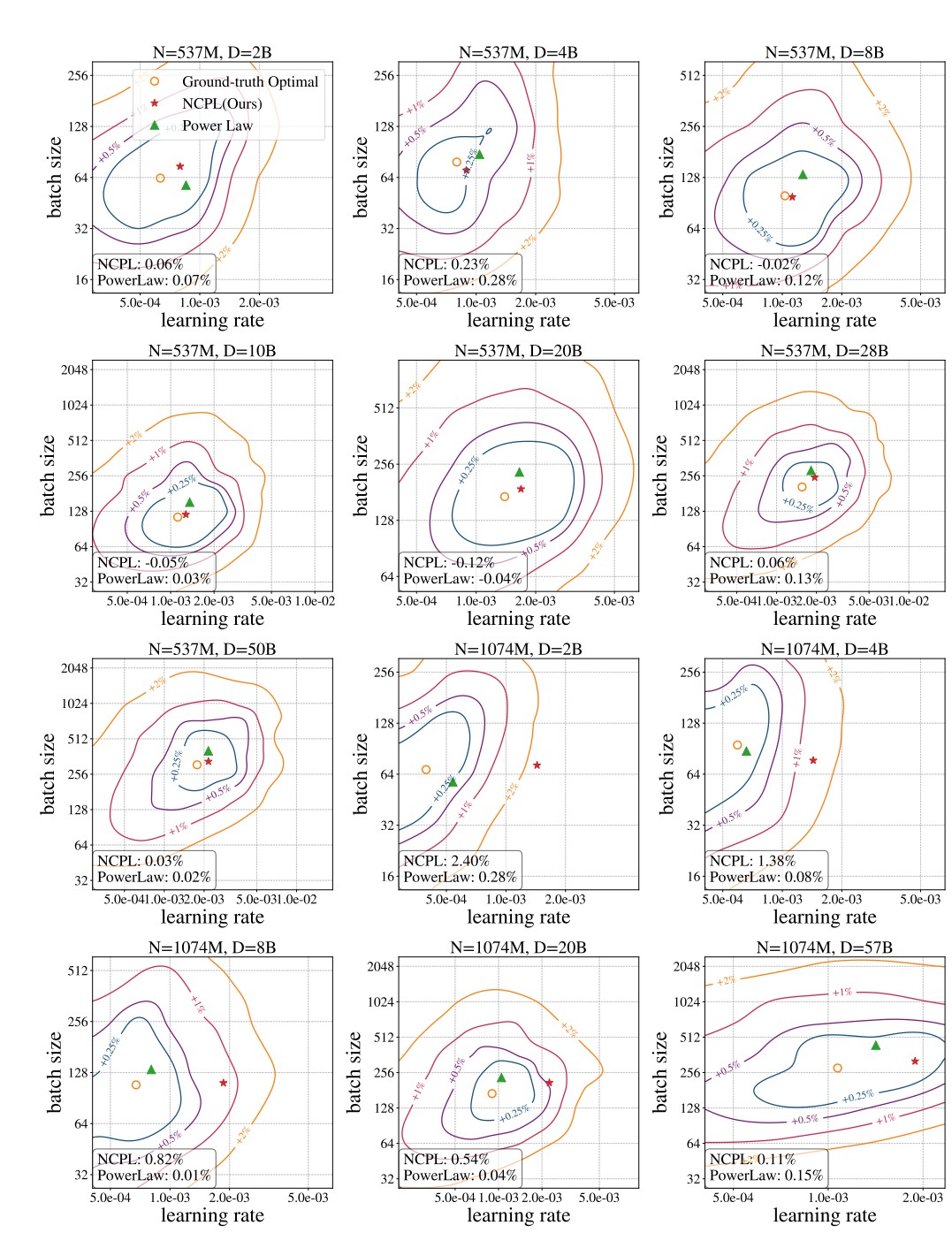

Figure 10: Predicted optimal learning rates and batch sizes for NCPL and the power law baseline on all 12 held-out OOD $(N, D)$ pairs, together with their relative loss.

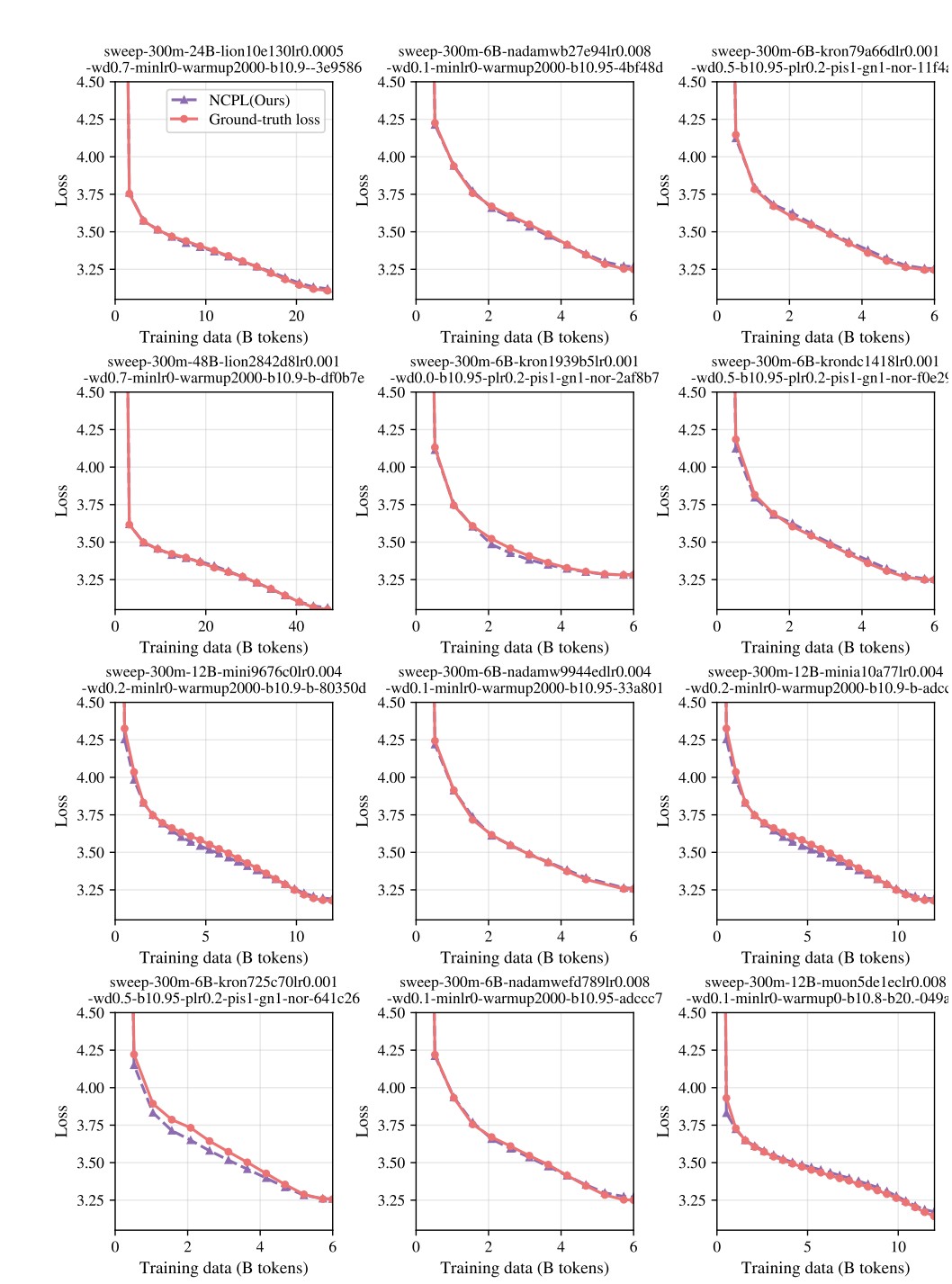

Figure 11: Loss-curve prediction on 12 randomly sampled ID runs (Marin, 300M). Each subplot title is the corresponding run name from the original Wandb project (Wen et al., 2025).

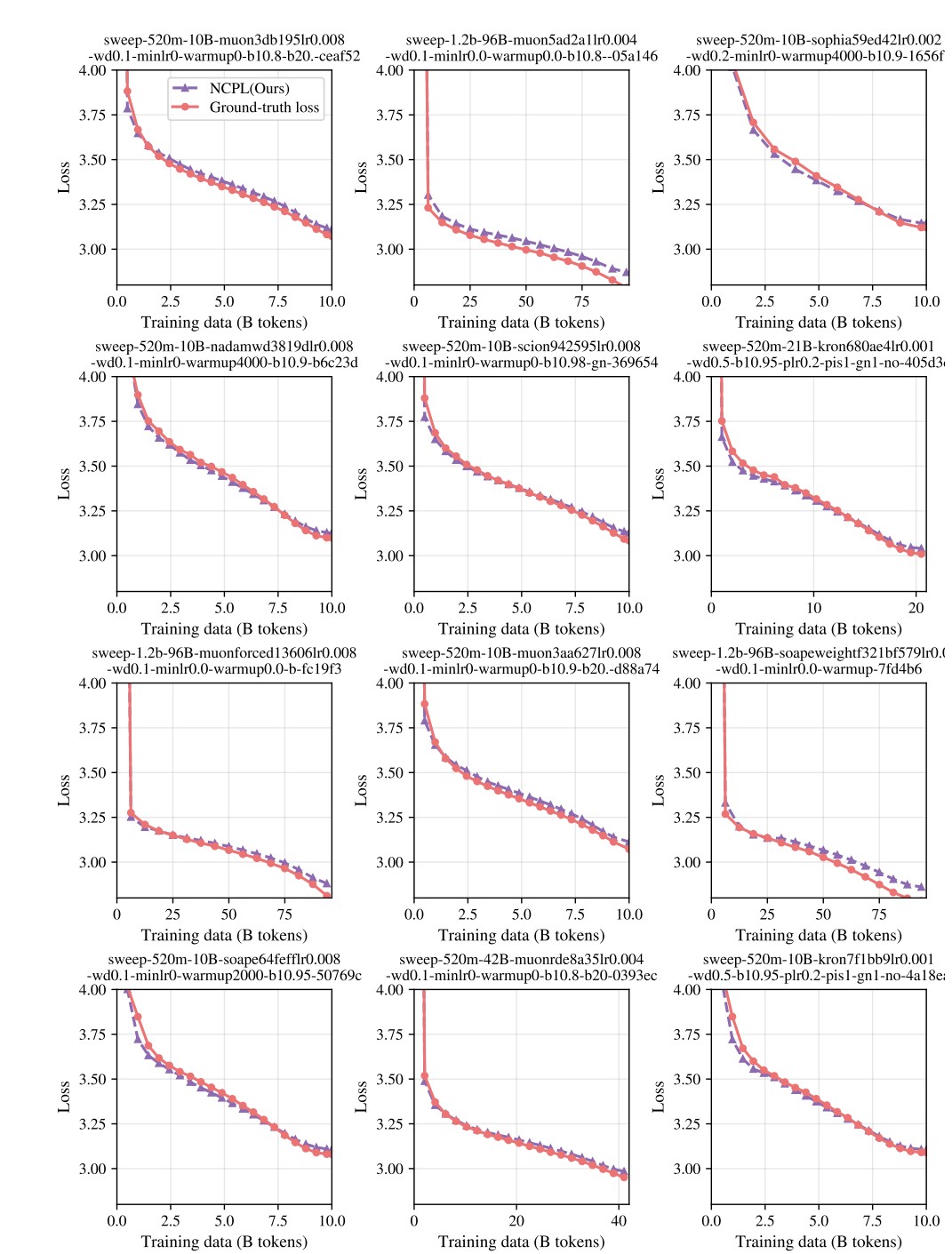

Figure 12: Loss-curve prediction on 12 randomly sampled OOD runs (Marin). Each subplot title is the corresponding run name from the original Wandb project (Wen et al., 2025).

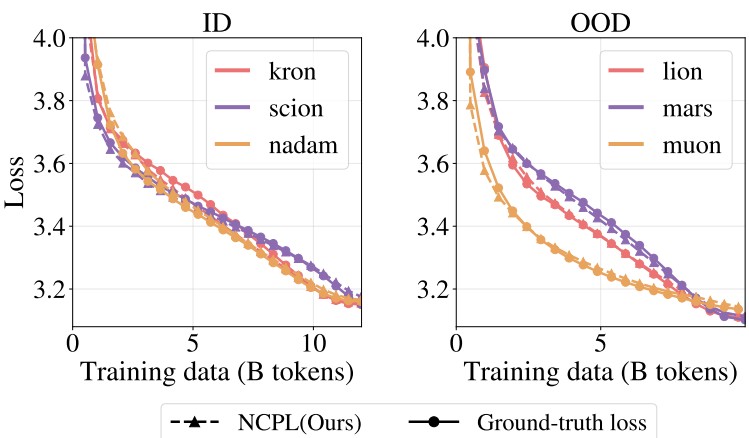

Figure 13: **Loss curve prediction.** Ground-truth and predicted pretraining loss curves under different optimizer settings on the ID and OOD validation sets. NCPL closely tracks the overall trajectories and captures optimizer-specific curve shapes. Setting: Marin Dataset. **Left:** ID validation, $N = 300M$, $D = 12B$, optimizers Kron/Scion/Nadam. **Right:** OOD validation, $N = 520M$, $D = 10B$, optimizers Lion/Mars/Muon. Results of loss curves under different hyperparameter are shown in fig. 4.

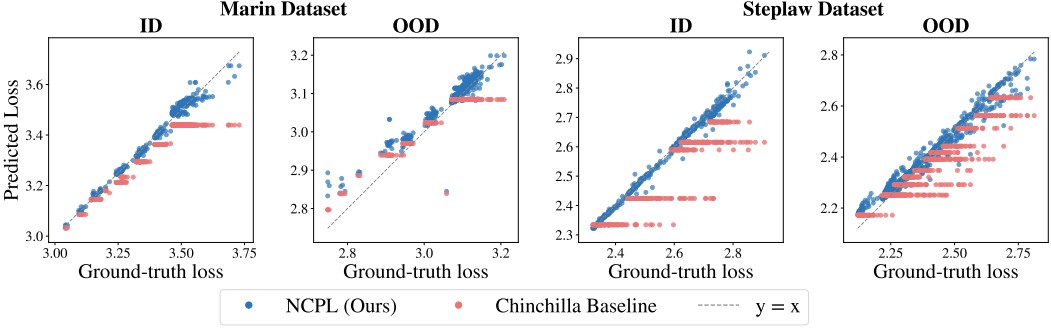

Figure 14: Predicted vs. ground-truth final pretraining loss of NCPL trained from scratch (1.7B model).

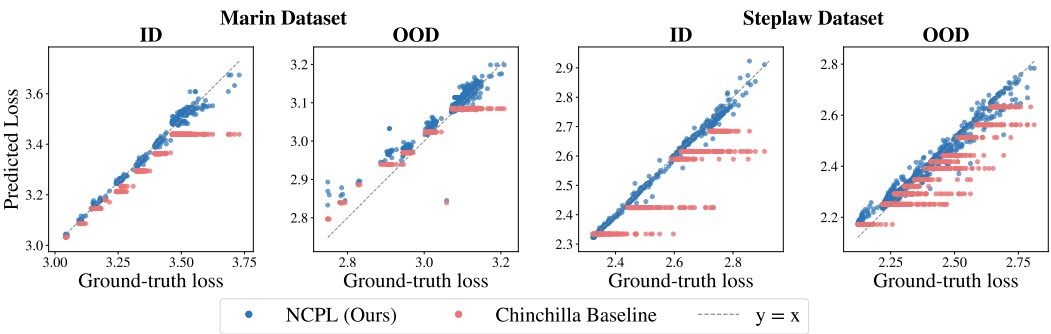

Figure 15: Predicted vs. ground-truth final pretraining loss of NCPL traineds from scratch (135M model).

