# OpenReview forum: "Configuration-to-Performance Scaling Law with Neural Ansatz"
_ICLR.cc/2026/Workshop/Sci4DL — Sci4DL 2026_

### Official Review · Reviewer_XUYe · 2026-02-11

**Fit:** 2
**Significance:** 2
**Confidence:** 3

**Summary:**

The paper proposes a Neural Configuration-to-Performance Scaling Law (NCPL), a LLM-based regressor that can predict LLM pretraining perfomance given pretraining configuration information. In contrast to closed-form functional forms that must be pre-specified, NCPL could more efficiently and reliably capture complex and nonlinear interactions between hyperparameters.

**NCPL model**
In the main paper, NCPL is a pretained LLM (Qwen3-1.7B) finetuned with additional numerical tokens to predict the residual loss relative to a fitted Chinchilla-law baseline. Ablations for these design choices (numerical tokens and residual prediction) are included in the appendix, as well as ablations for using a smaller 135M model and for training from scratch instead of finetuning.

**Data for training and evaluation**
Open-source training logs are used to create the Marin and Steplaw datasets, each with ~2.5k training runs, for training and evaluating NCPL. Training runs for larger models (>430M) are completely withheld for OOD validation, and 20% of remaining runs are kept as a withheld ID validation set.

Significant differences between the two datasets complicate interpretation of the results. In Steplaw, configuration variation is restricted to a small set of continuous factors ("N , D, learning rate, and batch size", L854). In Marin, many heterogeneous fields vary jointly ("e.g., optimizer choices and a wide range of hyperparameters", L856).
Furthermore, Steplaw performance is evaluated with smoothed training losses, while Marin is evaluated with C4 validation losses.

**Key results and contributions**
Reported results are positive, supporting the value and significance of NCPL as a novel method. NCPL more accurately predicts performance compared to a Chinchilla scaling law baseline which only takes into account model and data size, but not configuration information. NCPL also allows cost-effective joint hyperparameter tuning, and accurate loss curve prediction for 10-100B tokens across configurations in Marin (but not Steplaw?). The paper also shows that NCPL can qualitatively capture nuanced hyperparameter interactions, in this case specifically: weight decay and optimizer choice.

**Strengths:**

**Significance**
The problem addressed in this work is arguably significant, especially from an applied perspective where NCPL could support cost-effective hyperparameter optimization for LLM scaling experiments.
More generally, the curation of the Marin and Steplaw datasets supports future work on mapping training configurations to performance.
It was not clear to me from the paper how this work relates to (or could support) scientific understanding of DL, but I believe that understanding the complex and non-linear effects of configurations on performance is an important problem that is particularly relevant to this workshop and scientific understanding of DL, but unfortunately only briefly touched on in the paper.

**Quality**
Showing that LLMs can be trained to accurately predict LLM pretraining performance as a function of training configurations is perhaps not surprising at this point, but definitely non-trivial to do well; and this paper includes reasonably comprehensive experiments and ablations to substantiate their results. Results are mostly presented in a transparent way that directly addresses potential issues rather than sweep them under the rug (e.g. discrepancies between Marin and Steplaw results in Table 2, or less positive results for Marin in Figure 6).

**Soundness**
While I believe the soundness of the results can definitely be improved in some cases, I also believe they are generally appropriate for workshop acceptance.

**Clarity**
The paper is well written with key ideas, both in terms of methods and results, clearly communicated and easy to parse.
I appreciate the time and effort that has clearly gone into making this paper clear and concise.

**Suggestions:**

**Relevance to workshop on scientific understanding**
Despite the strengths of the paper, I saw its topic and contribution more in terms of applied research to support practitioners in scaling experiments, rather than in terms of advancing or supporting scientific understanding of DL.
Therefore, I initially decided to reject the paper because I felt it was too poorly aligned with the workshop.

However, I thought more about it and decided that, even though the paper as written is arguably not aligned with the workshop, the general problem it implicitly addresses (capturing complex, non-linear interactions between hyperparameters and their effect on LLM pretraining performance) is actually a very good fit for this workshop. Therefore, my first suggestion if the paper is accepted would be to make this connection more explicit in the paper and try to include additional experiments (or even just discussion) that more directly relate NCPL to scientific understanding of DL.


**Baseline comparisons that better support motivating hypothesis**
One of the main results and stated contributions is that NCPL can accurately predict LLM performance across configurations, particularly in comparison with Chinchilla scaling laws (Fig 1a,b).

However, Chinchilla scaling laws inherently do not take configurations into account, and so to me this comparison is neither fair nor meaningful. I would much rather see comparisons with existing functional forms that take into account different hyperparameters, but that do not cover the full configuration, or that fail to capture complex and non-linear interactions between hyperparameters.
Addressing these issues seems to be the main motivating hypothesis for NCPL, and using only Chinchilla scaling laws as a baseline does not support it.

Results for hyperparameter selection already include such a hyperparameter-aware power law as a baseline (i.e. Figs 1c, 2, and 9-10).
I would strongly suggest to include these as baselines throughout other experiments to more directly support your motivating hypothesis.
In my opinion, even negative results (e.g. Fig 9-10 where the power law baseline is consistently better than NCPL for larger 1B models) would not invalidate the paper and actually make it stronger for this workshop if appropriately analyzed and discussed.
I fully expect NCPL to improve performance prediction in some cases and degrade it in others.
Understanding what these cases are is more valuable to me than trying to show, maybe falsely, that NCPL is consistently better (both as a practitioner who might want to use NCPL for scaling experiments, and as a researcher who might want to build on NCPL towards better scientific understanding of DL).

**Statistical information that make results less ambiguous or potentially misleading**
While experiments are reasonably comprehensive, the quantitative results often cannot be reliably interpreted without more statistical information.

For example, results from Table 2 are sometimes conflicting in a way that cannot be disambiguated without e.g. standard deviations across combinations of (N,D) or other configuration hyperparameters.

Similarly, the figures for hyperparameter selection (Figs 7-10) show results individually for a small number of (N,D) pairs.
While these are described as representing the entirety of validation sets, I would like to see statistical information that quantitatively summarizes how well NCPL performs compared to a power law baseline across different configurations.
To what extent these results are specific to or generalize to certain configurations is not possible to determine from the current results.

**Evaluating across datasets more consistently**
Performance is measured differently for the Marin and Steplaw datasets; using C4 validation loss for one, and smoothed training loss for another. The justification for (and effect of) this decision is not addressed adequately, as far as I can tell, beyond a brief comment on variance in training loss.

On one hand, this is problematic because it's not clear what practitioners adopting NCPL should do in what contexts.
On the other, it's not clear to what extent different results are specific to this preprocessing decision or generalize more broadly.
I would consider adding an ablation or at least further discussing why you chose two different approaches for Marin and Steplaw, what differences (if any) this introduces in the datasets, and how this might affect results.

**Reporting across datasets more consistently**
A related issue is the inconsistency with which different results are reported for different datasets.

I appreciate e.g. [Figs 11,12] that explicitly address potential issues with cherry-picking, or [Figs 14,15] that show Marin and Steplaw results side-by-side despite more negative results for Marin OOD.

However, most of the paper seems to selectively report results on different datasets for different evaluations.
Given the transparency I saw throughout the paper, I doubt this is deliberate cherry picking of results, but I would strongly suggest more consistently reporting results across datasets. I found that, in cases were results were reported and compared for both datasets, the discussion on what underlies differences between datasets to be particularly valuable.


**Formatting and typos:**
- L091 "finetuing" -> "finetuning"
- “in Stage 1” -> "Stage 1"
- L191 “often prefer" -> “can benefit from”
- L729 broken ref/hyperlink

---

### Official Review · Reviewer_iz3u · 2026-02-16

**Fit:** 3
**Significance:** 2
**Confidence:** 2

**Summary:**

The paper proposes NCPL, a Neural Configuration-to-Performance Scaling Law that predicts final pretraining loss from a training configuration. NCPL is implemented as an LLM-based predictor trained on pretraining logs from two open-source projects, and it reports lower prediction error than configuration-agnostic baselines such as the Chinchilla scaling law. The model is also positioned as enabling joint tuning of multiple hyperparameters beyond model and dataset size.

**Strengths:**

1. The paper is well written and easy to follow. Many questions are addressed directly in the main text, and additional details in the appendix resolve most remaining ambiguities.
2. The core idea is conceptually simple yet interesting, and appears effective in practice.
3. The experimental results indicate that NCPL improves over the considered baselines, suggesting stronger predictive accuracy for estimating pretraining loss from training configurations.

**Suggestions:**

While NCPL predicts pretraining loss from training configurations (including model and dataset size), I would be cautious in framing it as a scaling law without further clarification. Traditional scaling laws are typically compact and interpretable, whereas NCPL appears closer to a learned meta-model that maps configurations to loss. It would strengthen the paper to clarify in what sense NCPL extends the notion of scaling laws (e.g., whether it recovers known scaling trends, enforces monotonicity with compute, or yields interpretable structure), and to better align the terminology with the method’s properties. The existing experiments in the paper show the feasibility of training a parametrized model to predict loss values and curves from training configurations. However, the practical adoption of this model as a scaling law model is yet to be clarified and discussed.

A second suggestion concerns transfer across datasets. The paper reports results separately for each dataset/source of logs. If the learned relationships between hyperparameters and loss are intended to be universal, it would be valuable to include a cross-dataset transfer evaluation (e.g., train on one source and test on the other, or hold out an entire dataset/source). If such transfer is not expected or not meaningful due to dataset-specific effects, the paper should motivate this explicitly and clarify the intended scope of generalization.

---

### Official Review · Reviewer_v3ff · 2026-02-24

**Fit:** 3
**Significance:** 2
**Confidence:** 3

**Summary:**

In this paper authors propose a novel way to fit a scaling law for language model training: they use a pre-trained LLM (Qwen3 1.7B base model) to map a configuration of training setup to predict validation loss and optimal hyperparameters for higher scales. They show that their method has better generalization than Chinchilla scaling law for the final loss prediction.

**Strengths:**

- The proposed method can be used for both validation loss and optimal hyperparameters prediction.
- The method generalizes better than Chinchilla for unseen model scales and can be used for the training curve estimation.

**Suggestions:**

The baselines that were used in the paper - Chinchilla and StepLaw - have incomparably less number of parameters than the model that was used in this paper. For example, Chinchilla has only 5 parameters that are used for the fit while the model that is used in this work is not only pre-trained but has 1.7B parameters. StepLaw has 3 parameters for optimal learning rate and 2 for optimal batch size estimation. Despite this, for optimal hyperparameters estimation (fig. 2) NCPL doesn't even show significantly better accuracy than StepLaw. So it's not clear whether this method is ultimately better considering the overall complexity of the model that is used for prediction.

---

### Meta-Review · Area_Chair_9bck · 2026-03-02

**Recommendation:** Accept

**Metareview:**

The paper proposes NCPL, a LLM-based regressor that can predict LLM pretraining perfomance, it could support efficient scaling. It would be great if the authors could clarify on what sense NCPL extends the notion of scaling laws. I see this is a good fit and the paper is well written. I recommend an accept.

---

### Decision · Program_Chairs · 2026-03-02

Accept